# COVID-19 Vaccination Willingness and Vaccine Uptake among Healthcare Workers: A Single-Center Experience

**DOI:** 10.3390/vaccines10040500

**Published:** 2022-03-24

**Authors:** Marija Zdravkovic, Viseslav Popadic, Vladimir Nikolic, Slobodan Klasnja, Milica Brajkovic, Andrea Manojlovic, Novica Nikolic, Ljiljana Markovic-Denic

**Affiliations:** 1University Clinical Hospital Center Bezanijska Kosa, Faculty of Medicine, University of Belgrade, 11 000 Belgrade, Serbia; sekcija.kardioloska@gmail.com (M.Z.); viseslavpopadic@gmail.com (V.P.); slobodan.klasnja@gmail.com (S.K.); brajkovic.milica@yahoo.com (M.B.); andrea.m93@gmail.com (A.M.); novica.nikolic87@yahoo.com (N.N.); 2Faculty of Medicine, University of Belgrade, 11 000 Belgrade, Serbia; 3Faculty of Medicine, Institute of Epidemiology, University of Belgrade, 11 000 Belgrade, Serbia; nikolicvladimir32@gmail.com

**Keywords:** COVID-19, vaccination, attitudes, healthcare workers

## Abstract

Healthcare workers (HCWs) are at higher risk of developing COVID-19 due to their professional exposition to the SARS-CoV-2 virus. This study assesses the intention of vaccination against COVID-19 before the vaccines were approved, and the rate of vaccine uptake during the first nine months of immunization among HCWs. A cross-sectional seroprevalence study was carried out during July 2020 in University Clinical Hospital Center Bezanijska Kosa in Belgrade, Serbia that included 62.8% of all HCWs. Besides serological testing for IgG antibodies, data about HCWs’ intention to accept COVID-19 vaccination if a vaccine became available were collected. This cohort of HCWs was followed up until the end of October 2021 to assess the number of vaccinated and PCR-positive staff. In the cross-sectional study, 18.3% HCWs had positive SARS-CoV-2 IgG antibodies without difference with IgG-negative HCWs regarding age, gender, profession type, and years of service. Before vaccines became available, a significantly higher percentage of IgG-positive HCWs compared to IgG-negative HCWs was unsure whether to be vaccinated (62.5% vs. 49.0%), and significantly fewer stated that they would not be vaccinated (16.7% vs. 25.1%). When the vaccines became available in Serbia, among IgG-negative HCWs, those who stated clear positive (yes) and clear negative (no) attitude toward vaccination before the immunization period had begun were vaccinated at 28% and 20%, respectively, while 51% of unsure HCWs received a vaccine (*p* = 0.006). Among IgG-positive HCWs, there was no statistical difference in vaccine uptake regarding those with previous negative, positive, and unsure opinions about vaccination (*p* = 0.498). In multivariate analysis, independent factors associated with uptake were being female (OR = 1.92; 95%CI: 1.04–3.55), age of 30–59 years, previously vaccine-unsure (OR = 1.84; 95%CI: 1.04–3.25), and those with previous positive vaccine attitudes (OR = 2.48; 95%CI:1.23–5.01), while nurses were less likely to become vaccinated (OR = 0.39 95% CI: 0.20–0.75) These findings indicate a positive change in attitudes of HCWs towards COVID-19 vaccination.

## 1. Introduction

Healthcare workers (HCWs) are at higher risk of becoming infected and developing COVID-19 due to their professional close and long contact with patients and exposure to the SARS-CoV-2 virus. According to WHO data, between 80,000 and 180,000 HCWs died from COVID-19 between January 2020 and May 2021 [1]. HCWs are at seven times higher risk for COVID-19 than that of other employers [2]. This profession is not only related to the risk of COVID -19 diseases, but can also cause high levels of stress and psychological burden, especially in intensive-care-unit HCWs [3]. According to the conclusions of recently published systematic reviews and meta-analyses of 28 studies from different countries, more than 50% of HCWs became infected with SARS-CoV-2 [4]. Therefore, nonpharmaceutical measures, particularly wearing personal protective equipment (PPE), are important in disease prevention among health professionals. Nevertheless, immunization is the most cost-effective preventive measure for reducing the spread of infectious agents for the general population and individual protection [5]. Given that HCWs are a particularly vulnerable group during the COVID-19 pandemic, there are opinions that COVID-19 vaccine should be mandatory and not just recommended for them [6].

The vaccination of health workers is of twofold importance. On the one hand, they play a key role in raising public awareness of the importance of vaccination. On the other hand, vaccinated HCWs benefit from relieving individual symptoms and severe disease, and reduce the transmission of infections to patients. Even if vaccinated people transmit the virus like unvaccinated people, they can clear the virus faster, meaning that SARS-CoV-2 is present for less time in the upper nasopharynx of vaccinated people [7].

Health professionals are often expected to have a positive attitude towards vaccination, but this is not always the case. A positive attitude towards vaccination against COVID-19 was in the range of 60–90% among physicians and about 40–60% among nurses in the prevaccine era [8,9,10]. A higher proportion of HCWs were hesitant about vaccination than those who did not want to be vaccinated according to a study conducted in France, and French-speaking parts of Belgium and Canada [11]. They represent an important group whose change of opinion from hesitance to willingness to be vaccinated would increase vaccine coverage [12].

Recognizing the importance of vaccination in disease prevention, Serbia was the first country in the region that ordered more types of vaccines from different manufacturers and the World Health Organization (WHO) COVAX program [13]. On 24 December 2020, the first doses of the vaccine were administered, and a broad national vaccination program began in January 2021 [14]. This was preceded by a massive mass media campaign, and engaging the public on acceptance and vaccine uptake. Training and support for medical workers to deliver the new vaccines were also provided. In addition to traditional vaccination points and primary healthcare centers, new fully equipped vaccination points were established, such as at fairs. According to the recommendation of the National Immunization Technical Advisory Group of Serbia, HCWs belonged to the priority vaccination group.

The aims of this study were: (a) assessing the intention of vaccination against COVID-19 before vaccines were approved; (b) assessing the actual uptake of vaccination; and (c) investigating factors associated with vaccine uptake after its implementation among HCWs.

## 2. Materials and Methods

### 2.1. Study Setting and Population

A cross-sectional seroprevalence study was carried out at the end of June 2020 in University Clinical Hospital Center Bezanijska Kosa in Belgrade, Serbia. The hospital was then in a non-COVID-19 regime. All presently working HCWs were invited to participate in the study without exclusion criteria. Out of 916 employees working at the hospital at the time of the study, 575 entered the study voluntarily. Written informed consent was obtained, and participants donated blood for serological testing for IgG antibodies. The infection-control staff administered the questionnaire to people involved in the study. During the entire time of filling in the questionnaire, the staff was available for all questions and concerns.

Besides sociodemographic data about age, sex, occupation, and length of employment, they answered questions “will you be vaccinated if a vaccine against COVID-19 would be available” and “do you think that there is a great likelihood of catching COVID-19 if you would not be vaccinated?” Possible responses for both questions were “no”, “not sure”, and “yes”. Additional analysis showed that the group unwilling to be vaccinated against COVID-19 and the vaccine-unsure group were not homogeneous. Therefore, the two above-mentioned variables (questions) were split into three categories: no, unsure, and yes.

When the respondents had finished filling out the questionnaire, the trained nurse collected about 5 mL of venous blood under aseptic precautions for serological analysis. The determination of IgG antibodies against SARS-CoV-2 was performed using indirect ELISA SARS-CoV-2 IgG. The test was created at the Institute for Application for Nuclear Energy (INEP, Belgrade, Serbia), and the was CE labeled and registered as the IVD at the Serbian Agency for Drugs and Medical Devices (ALIMS). Results are expressed semiquantitatively as a relative index to the reference calibrator sample. Cut-off values for IgG were established on the basis of ROC curve analysis. For antibody indices higher than 20, the IgG test was considered to be positive.

A cross-sectional study was converted into a prospective cohort study. The cohort of HCWs that entered the seroprevalence study was followed for nine months from January to the end of September 2021.

The endpoint of this study was willingness to be vaccinated before the vaccination was available, and later COVID-19 vaccine uptake.

During the following period, the hospital was transformed into a COVID-19-dedicated hospital and remained in that status until May 2021. Due to the smaller number of COVID-19 patients in the country during the summer months, the hospital was only not in the COVID-19 regime in the period of July–August. In September, due to the increased number of patients, the hospital was retransformed into a COVID-19-dedicated facility. Since January 2021, vaccination has been organized for all staff of that hospital.

Serbia re-engineered the vaccine approval process by building AI-powered software that sped up each phase three times while taking citizens’ preferences into account. The country also implemented a new digital platform to schedule vaccine deliveries called the System for Immunization Management of the Republic of Serbia, which enables the real-time monitoring of the immunization of each citizen. Serbia adopted an approval mechanism for COVID-19 vaccinations that gives citizens the option to choose which one of the offered vaccines they preferred and in which location they wanted to be vaccinated. At the hospital level, a specific place and time for the vaccination of employees were organized. If one of the employees was vaccinated outside of that, they were obliged to show a certificate of vaccination to the infection-control team responsible for collecting data from vaccinated persons. In this way, there was double control of the data on vaccinated HCWs.

An HCW was considered to have been tested for COVID-19 if they had had SARS-CoV-2 RT-PCR at least once regardless of the reason for testing. Diagnosis of COVID-19 was based on laboratory-confirmed COVID-19 infection or a clinical diagnosis of COVID-19 according to imaging techniques. Data of RT-PCR testing results from some referent virology laboratories in the city were obligatorily entered into the electronic system of the national COVID-19 database. Each RT-PCR-tested employee is obliged to submit the test data to the occupational health department (OHD) of the hospital and the infection-control team, which can check the data in the national COVID-19 electronic database using the employee’s personal identification number. In addition, the OHD is in charge of keeping records of hospitalized HCWs. The infection-control team contacts all hospitalized HCWs or their relatives for further surveillance details. Thus, variable hospitalization was dichotomous (yes/no) for all HCWs.

### 2.2. Statistical Analysis

Descriptive and analytical statistical methods were used in data processing. Data are presented as mean ± SD and number (percentage) for categorical variables. A chi-squared test was used to analyze categorical data, and an independent t-test was used for continuous variables. Univariate logistic regression analyses were performed to identify predictors for vaccination against COVID-19 as an outcome. The multivariable logistic regression model with backward Wald as a method was performed to identify independent factors associated with vaccination against COVID-19. Variables that were significantly associated with these outcomes at significance level <0.1 in univariate logistic regression analysis were entered into the multivariable logistic regression model. Odds ratios (OR) with 95% confidence intervals (CIs) were computed, and the Hosmer–Lemeshow goodness-of-fit test was performed to assess overall model fit. Statistical analysis was performed using SPSS version 26.0 software (SPSS Inc., Chicago, IL, USA).

### 2.3. Ethics

The study was organized according to the principles of the Declaration of Helsinki of 1975, as revised in 2008, and approved by the ethics committee in the hospital.

## 3. Results

Out of all healthcare workers in the hospital, 575 (62.8%) decided to participate in the cross-sectional study. The demographic characteristics of participants are shown in Table 1. Of all participants, 18.3% (105/575) had positive SARS-CoV-2 IgG antibodies. The mean age of included HCWs was 41.6 years; the majority were female (80.3%), and 48.2% were nurses. There was no statistically significant difference among IgG-positive and IgG-negative participants regarding age, gender, profession type, and years of service (Table 1). Of the participants, 93% responded to the question about possible vaccination when a vaccine is available. About half of the respondents (51.4%) were unsure if they would like to be vaccinated against COVID-19, while 23.7% had a negative, and 24.9% had a positive attitude toward vaccination. Among IgG-positive HCWs compared to IgG-negative HCWs, there was a significantly higher percentage of those who were unsure about vaccination, and a lower percentage of those who stated that they did not want to be vaccinated (62.5% vs. 49.0% and 16.7% vs. 25.1%, respectively) when vaccines became available in Serbia.

One-fifth of the respondents (20.7%) thought that there was a great likelihood of catching COVID-19 if they would not be vaccinated, while 41.4% of them were uncertain about this issue without a difference regarding serological status (Table 1).

During the first nine months 2021, 66.6% (383/575) of participants were vaccinated with one of four COVID-19 vaccines available in Serbia from the end of 2020: 81.2% (311/383) were IgG-antibody-negative and 18.8% (72/383) were IgG-antibody-positive HCWs at the cross-sectional study. Of our cohort, 47.3% received the Sinopharm (Beijing) BBIBP-CorV (Vero Cells) vaccine, 38.9% the Pfizer–BioNTech (Comirnaty) vaccine, and 13.8% the Sputnik V (Gam-COVID-Vac) vaccine (Figure 1). Nobody received the AstraZeneca (Vaxzevria) vaccine. Among profession type, 73.8% of physicians, 61.7% of nurses, and 69% of other staff received a vaccine. A significantly higher percentage of physicians received the Pfizer–BioNTech (Comirnaty) vaccine than that of nurses and other staff, while more other staff received the Sinopharm (BBIBP-CorV) vaccine compared to nurses and physicians (*p* < 0.001) (Figure 1).

Out of 403 HCWs who were vaccine-unsure or who did not intend to accept the COVID-19 vaccine before the beginning of COVID-19 immunization, 64.3% were vaccinated during 2021 (69.2% of vaccine-unsure, and 53.5% of those who had declared that they did not want to be vaccinated). Among IgG-negative HCWs, only 28% and 20% of those who stated clear positive (yes) and clear negative (no) attitude toward vaccination before the immunization period had begun were vaccinated, respectively, while 51% of unsure HCWs received a vaccine (*p* = 0.006). There was no significant difference in vaccination rate according to the previous attitude toward vaccination among IgG-positive HCWs (*p* = 0.498) (Table 2).

Predictors of healthcare workers’ COVID-19 vaccine uptake are presented in Table 3. In univariate analysis, COVID-19 vaccine uptake was associated with the female sex, age of 30–59 years, >20 years of service, SARS-CoV-2 PCR test performed, hospitalization, unsure and positive attitudes about vaccination, and nurses were less likely to be vaccinated. Multivariate analysis shows that females are more likely to be vaccinated than males are. Age of 30–59 years was associated with a higher uptake of COVID-19 vaccine. Nurses were less likely to be vaccinated, confirming the results from univariate analysis. After multivariate analysis, HCWs who were unsure about vaccination had almost twice higher odds (OR: 1.84; 95%CI: 1.04–3.25) concerning those with a negative attitude, while HCWs with a previous positive attitude had 2.48 higher odds (OR: 2.48; 95%CI: 1.23–5.01) for vaccination.

## 4. Discussion

This study showed that, before vaccines were approved, less than one-quarter of participating HCWs had reported willingness to be vaccinated against COVID-19, with a higher percentage of negative SARS-CoV2 IgG antibodies. In a study conducted in France during a similar period, 77% of participants expressed their willingness to be vaccinated if a vaccine against COVID-19 was available [9]. In Switzerland, the percentage of staff intending to accept vaccination was lower, less than half of participants, while about 30% were unsure [15]. A recently published systematic review and meta-analysis of cross-sectional studies revealed that overall acceptance of COVID-19 vaccination was 51% [16]. Intriguingly, this percentage is lower than that among the general population [17]. It would be expected that health workers would express higher intention to be vaccinated because they are in the front line with COVID patients and are much more medically knowledgeable than the general population. However, they may be more concerned about vaccine effectiveness and safety, which may alter their opinion about vaccine uptake [18]. Results of the study conducted among HCWs in the United States showed that almost half of about 3500 respondents thought that COVID-19 vaccination should be voluntary [17]. The discussion on whether the introduction of mandatory vaccination is ethical is still open [6].

The higher percentage of unsure HCWs with positive IgG antibodies found in our study may be explained by the lack of knowledge at the time when the cross-sectional study was conducted regarding possible reinfection and the need to vaccinate persons who had already had COVID-19.

Repeated cross-sectional surveys in Hong Kong during the first and third waves of the local COVID-19 epidemic that included different job professions showed increased willingness for COVID-19 vaccination during the three waves of the current pandemic [19]. The proportion of COVID-19 vaccine acceptance among HCWs varied from 27.7% to 77.3%, and was different in different regions of the same country [20]. Our study found that more than two-thirds of vaccine-unsure respondents and half of those who had had a negative attitude towards the vaccine at the beginning of the study were vaccinated when the vaccine became available in Serbia. A possible explanation is that our hospital was transformed into a COVID-19-dedicated hospital after a cross-sectional study, and that daily care and treatment of COVID-19 patients raised awareness of the importance of preventive measures, especially vaccines. Fortunately, four types of vaccines have been available in Serbia since the beginning of 2021, and employees were able to decide for themselves which vaccine they want to receive. Physicians preferred the messenger RNA (mRNA) vaccine, probably due to better knowledge of the new manufacturing technology compared to other hospital staff [21]. Other HCWs were more confident in vaccines produced using traditional technology, such as inactivated whole virus vaccines.

In our study, independent factors that influenced the COVID-19 vaccine acceptance were gender, older age, occupation, and unsure and positive attitudes about vaccination. Females were 1.8 times more likely to receive vaccination than males were, which is contrary to obtained findings in the systematic review and meta-analysis of cross-sectional studies among HCWs [16] and the general population [22]. Our results are in concordance with results reported from studies in Ghana and Saudi Arabia [23,24].

As in many studies [15,16,25], we also found that uptake of vaccine among HCWs was associated with age groups of 30–59 years. This finding can be explained by the fact that COVID-19 mortality is higher in the older population [26], which probably encouraged older adults to accept vaccination.

According to our results, nurses were less likely to be vaccinated against COVID-19, although they have longer and closer contact with patients [20]. These findings were confirmed in other studies.

In this study, HCWs who had a positive attitude towards COVID-19 vaccination before vaccines became available were vaccinated 2.5 times more than workers with a negative attitude. Further, a significant change of mind about vaccination among unsure HCWs was observed, i.e., those who had not yet decided to receive the COVID-19 vaccine were 1.8 times more vaccinated than those who were against vaccination. Indecisive people are a group that can change their minds if they obtain more information about the effectiveness of the vaccine, its safety, and the fact that, if vaccinated, it protects both themselves and those who may have wanted but could not be vaccinated against disease [27]. The speed of vaccine development, knowledge about vaccines, and the cultural context in society impact vaccination [28]. A steady increase in intention to receive a vaccine was observed among HCWs in the United States [29]. Intervention such as continuing education and awareness campaigns could increase the proportion of vaccinated HCWs [30].

The limitation of our study is that it was conducted in a single center using a cross-sectional design in the first part of the study. However, this hospital is a university facility that is composed of all departments except pediatrics. A large number of capital residents gravitate for treatment to it. The limitation of a cross-sectional study including sampling i.e., voluntary enrollment, may have resulted in selection bias. The majority of similar studies that revealed HCWs’ attitudes and willingness to accept COVID-19 vaccination were mainly carried on through online recruitment. The strength of our study is that it was conducted using paper questionnaires as a preferred way for data collection [31]. Completion of the questionnaire was organized in a comfortable room with a person who assisted respondents in filling it, which is a prerequisite for a high response rate.

## 5. Conclusions

The results of our study indicate a positive change in attitudes of HCWs towards COVID-19 vaccination. Independent predictors of HCW attitudes toward COVID-19 vaccination were the female sex, age >30 years, physician occupation, and unsure and positive attitudes about vaccination. The results of this and many similar studies are important, as they can affect the future willingness rate of HCWs and the overall population to become vaccinated against COVID-19.

## Figures and Tables

**Figure 1 vaccines-10-00500-f001:**
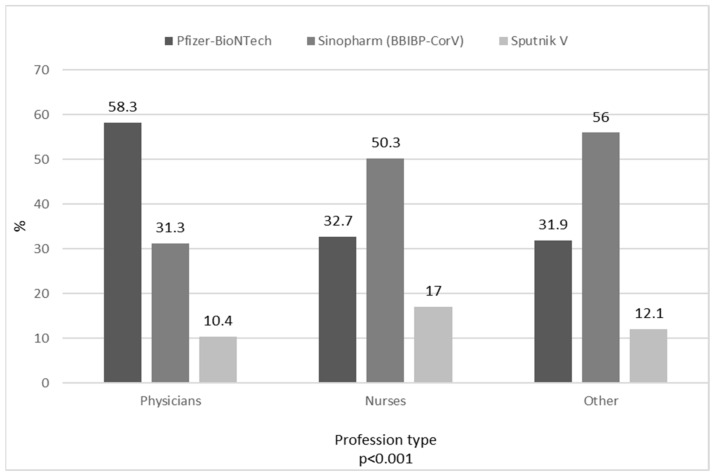
Vaccines administered to healthcare workers (vaccinated total, 383).

**Table 1 vaccines-10-00500-t001:** Demographic and SARS-CoV-2 testing characteristics.

	Total *n* (%)	SARS-CoV-2 Antibody IgG-Negative *n* (%)	SARS-CoV-2 Antibody IgG-Positive *n* (%)	*p* Value
Total	575 (100)	468 (81.7)	105 (18.3)	
Gender				
Male	113 (19.7)	88 (18.8)	25 (23.8)	0.244
Female	462 (80.3)	380 (81.2)	80 (76.2)	
Age (years)—mean ± SD	41.6 ± 11.3	41.5 ± 10.9	41.6 ± 11.3	0.517
18–29 years	110 (20.0)	91 (20.2)	18 (18.2)	
30–39 years	131 (23.8)	107 (23.8)	23 (23.2)	
40–49 years	149 (27.0)	123 (27.3)	26 (26.3)	0.917
50–59 years	141 (25.6)	112 (24.9)	29 (29.3)	
60–65 years	20 (3.6)	17 (3.8)	3 (3.0)	
Profession type				
Physicians	130 (22.6)	102 (21.8)	27 (25.7)	
Nurses	277 (48.2)	228 (48.7)	48 (45.7)	0.680
Other	168 (29.2)	138 (29.5)	30 (28.6)	
Years of service—mean ± SD	16.4 ± 11.6	16.5 ± 11.6	16.3 ± 11.9	0.850
0–20 years	255 (61.0)	206 (60.6)	47 (61.8)	0.840
21–40 years	163 (39.0)	134 (39.4)	29 (38.2)	
Will you be vaccinated?				
No	127 (23.7)	110 (25.1)	16 (16.7)	
Unsure	276 (51.4)	215 (49.0)	60 (62.5)	**0.050**
Yes	134 (24.9)	114 (26.0)	20 (20.8)	
Do you think that there is a great likelihood of catching COVID-19 if you would not be vaccinated?				
No	183 (33.9)	153 (34.6)	29 (30.2)	
Unsure	238 (44.1)	190 (43.0)	47 (49.0)	0.555
Yes	119 (22.0)	99 (22.4)	20 (20.8)	

SD—Standard deviation, bold: significant.

**Table 2 vaccines-10-00500-t002:** Vaccination status during the first nine months of 2021 in relation to attitudes toward vaccination and expected possibility to be infected with COVID-19 in the absence of vaccination.

Will You Be Vaccinated?	SARS-CoV-2 Antibody IgG Negative			SARS-CoV-2 Antibody IgG Positive			Total		
	Vaccination against COVID-19								
	No *n* (%)	Yes *n* (%)	*p* Value	No *n* (%)	Yes *n* (%)	*p* Value	No *n* (%)	Yes *n* (%)	*p* Value
No	52 (34.0)	58 (20.3)	**0.006**	6 (23.1)	10 (14.3)	0.498	58 (46.5)	68 (53.5)	**0.002**
Unsure	68 (44.4)	147 (51.4)		16 (61.5)	44 (62.9)		84 (30.8)	191 (69.2)	
Yes	33 (21.6)	81 (28.3)		4 (15.4)	16 (22.9)		37 (27.6)	97 (72.4)	
Do you think that there is a great chance that you will become sick from COVID-19 if you are not vaccinated?									
No	60 (39.5)	93 (32.1)	0.196	8 (30.8)	21 (30.0)	0.710	68 (37.7)	114 (62.3)	0.150
Unsure	64 (42.1)	126 (43.4)		14 (53.8)	33 (47.1)		78 (33.2)	159 (66.8)	
Yes	28 (18.4)	71 (24.5)		4 (15.4)	16 (22.9)		32 (26.9)	87 (73.1)	

Bold: significant.

**Table 3 vaccines-10-00500-t003:** Predictors of healthcare workers COVID-19 vaccine uptake: univariate and multivariate binary logistic regression analyses.

	Vaccinated against COVID-19 *n* (%)	Univariate Binary Logistic Regression OR (95% CI)	Multivariate Binary Logistic Regression OR (95% CI)
Total	383 (66.6)		
Gender			
Male	65 (57.5)		
Female	318 (68.8)	**1.63 (1.07–2.49)**	**1.92 (1.04–3.55)**
Age (years)—mean ± SD	43.3 ± 10.2		
18–29 years	52 (47.3)	ref.	ref.
30–39 years	81 (61.8)	**1.81 (1.08–3.02)**	**2.09 (1.08–4.05)**
40–49 years	120 (80.5)	**4.61 (2.66–8.01)**	**3.92 (1.97–7.83)**
50–59 years	115 (81.6)	**4.93 (2.80–8.70)**	**4.69 (2.27–9.71)**
60–65 years	13 (65.0)	2.07 (0.77–5.59)	1.04 (0.24–4.38)
Occupation			
Physicians	96 (73.8)	ref.	ref.
Nurses	171 (61.7)	**0.57 (0.36–0.90)**	**0.39 (0.20–0.75)**
Other	116 (69.0)	0.79 (0.47–1.32)	0.51 (0.24–1.06)
Years of service—mean ± SD	18.4±11.3		
0–20 years	156 (61.2)		
21–40 years	129 (79.1)	**2.41 (1.53–3.79)**	
SARS-CoV-2 antibody IgG positive	72 (68.6)	1.10 (0.70–1.73)	
SARS-CoV-2 PCR test performed (total tests 487)	342 (70.2)	**2.70 (1.70–4.29)**	
History of COVID-19			
PCR test negative (total 319) PCR test positive, nonhospitalized (total 148) PCR test positive, hospitalized (total 20)	226 (70.8) 98 (66.2) 18 (90.0)	ref. 0.81 (0.53–1.23) 3.70 (0.84–16.28)	
Will you be vaccinated?			
No	68 (53.5)	ref.	ref.
Unsure	191 (69.2)	**1.95 (1.26–3.00)**	**1.84 (1.04–3.25)**
Yes	97 (72.4)	**2.27 (1.36–3.81)**	**2.48 (1.23–5.01)**
Do you think that there is a great chance that you will become sick from COVID-19 if you are not vaccinated?			
No	114 (62.3)	ref.	
Unsure	159 (66.8)	1.22 (0.81–1.82)	
Yes	87 (73.1)	1.65 (0.99–2.72)	

SD, standard deviation; Bold: significant.

## Data Availability

Not applicable.

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
