# Peer review of "COVID-19 Vaccination Willingness and Vaccine Uptake among Healthcare Workers: A Single-Center Experience"

_vaccines, 2022, doi:10.3390/vaccines10040500_

Round 1

Reviewer 1 Report

This manuscript appears to be potentially very interesting and has significance. However, the use of English needs improvement in order to understand the key points being made. There are also several places where the wording is unclear. There is a need for extensive revision to ensure the reader understands to results and their interpretation.

The introduction should also provide some additional background about the area directly related to the current study. It appears that the study has more of historical perspective which includes attitudes prior to development of vaccines and what happened after they were available. This would help provide some perspective to the scope of the study.

The Y and X axes in figure 1 should be labeled. It is also not clear what the P<0.05 is referring to. Are there significant differences between the groups? If so, these differences should be clearly identified.

The results section should be clearer. There are statements made about predictors, but it is not clear what is the prediction. For example, it appears that the authors are stating that females would more likely get vaccinated than males, but this is not stated in that way. Again, this appears to be a result of a need for a revision to improve English usage.

Overall, this seems to have potential to demonstrating the change in attitude toward being vaccinated by health care workers after the development of the new covid vaccines.

Author Response

Response to Reviewer 1 Comments

Reviewer #1: This manuscript appears to be potentially very interesting and has significance. Point 1.However, the use of English needs improvement in order to understand the key points being made. There are also several places where the wording is unclear.

Response 1: We thank reviewer for this comment and tried our best to improve the English.

There is a need for extensive revision to ensure the reader understands to results and their interpretation.

Point 2. The introduction should also provide some additional background about the area directly related to the current study. It appears that the study has more of historical perspective which includes attitudes prior to development of vaccines and what happened after they were available. This would help provide some perspective to the scope of the study.

Response 2: We thank the reviewer. We deleted paragraph at lines 48-49 which was out of the scope of the paper and added three new paragraphs (paragraphs: 2-4) with additional information about vaccination attitudes and the introduction of COVID-19 vaccination in Serbia.

Point 3.The Y and X axes in figure 1 should be labeled. It is also not clear what the P<0.05 is referring to. Are there significant differences between the groups? If so, these differences should be clearly identified.

Response 3 : Thank you for this suggestion. We labeled X and Y axes in Figure 1. Also, we added p value for each vaccine. There is a significant difference for Pfizer-BioNTech eg. Significantly higher percentage of physicians received this vaccine (p<0.001) while higher proportion of other staff received Sinopharm (Beijing) BBIBP-CorV (Vero Cells) vaccines (p=0.001).

Point 4. The results section should be clearer. There are statements made about predictors, but it is not clear what is the prediction. For example, it appears that the authors are stating that females would more likely get vaccinated than males, but this is not stated in that way.

Response 4 : Thank you for this comment. We changed the section about predictors as follows “Predictors of healthcare workers' COVID-19 vaccine uptake are presented in table 3. In the univariate analysis, COVID-19 vaccine uptake was associated with female sex, age 30-59 years, years of service > 20 years, SARS-CoV-2 PCR test performed, hospitalization, hesitant and positive attitude about vaccination, while nurses were less likely to get vaccinated. Multivariate analysis shows that females would more likely get vaccinated than males. Age 30-59 years were associated with higher uptake of COVID-19 vaccine. Nurses were less likely to get vaccinated, confirming the results from the univariate analysis. After multivariate analysis HCWs who were hesitant about vaccination had almost two times higher odds (OR: 1.84; 95%CI: 1.04-3.25) concerning those with a negative attitude, while HCWS with a previous positive attitude had even 2.48 higher odds (OR:2.48; 95%CI:1.23-5.01) for vaccination.”

Overall, this seems to have potential to demonstrating the change in attitude toward being vaccinated by health care workers after the development of the new covid vaccines.

Reviewer 2 Report

The paper “COVID-19 vaccination wailings and vaccine uptake among 2 healthcare workers: a single-center experience” is a single-center study that addresses HCW’s attitudes toward COVID-19 vaccination and following uptake investigating predictive factors. The paper provides a piece of information about the topic that adds to the existing evidence but need to be amended in respect to several aspects, in particular concerning the methodology. My comments are as follows.

Title

  1. There is a typo in the title (wailings instead of willingness).

Abstract

  1. The results about the two groups being compared in respect to attitudes and following uptake of vaccination are not clearly reported.

Introduction

  1. The paragraph at lines 48-56 seems out of the scope of the paper. The authors should address the relevance of assessing vaccination attitudes and then describe the steps of the introduction of vaccination in Serbia.
  2. In the objective I would not mention the 9 months of follow up, but I would generically say that the objective was to assess the actual uptake of vaccination after its implementation.

Methods

  1. Authors said that all employees, including both medical and non-medical staff, were invited to participate to the study but, from results, it appears that only healthcare workers participated to the study. Indeed, it does not make sense to say that the sub-cohort of HCWs were involved in the follow up study. Please clarify this point.
  2. Who did administer the questionnaire to people involved in the study? The information should be reported in methods and not at the end of discussion.
  3. The primary endpoint of the study is not clear. Indeed, it should be the willingness to vaccinate before the vaccination was available and the following uptake of vaccination.
  4. Methods and cut-off used to measure IG antibodies should be reported.
  5. The source of data on vaccination status is not declared and source of information about RT-PCR testing and COVID-19 diagnosis are not clearly reported and not considered in the results.
  6. In the statistical analysis, I would suggest Authors to better explain which kind of analyses they performed in respect to the different endpoints.

Results

  1. Authors quoted “There was no statistically significant difference among IgG positive and IgG negative participants regarding age, gender, profession type, and years of service (Table 1).” Nevertheless, p-values were not reported in the table and methods section did not anticipate the use of test suitable for addressing the differences between the two groups in respect to all the variable considered that were not all categorical.
  2. Authors quoted “almost half of the respondents (48.0%) were uncertain if they would like to be vaccinated against COVID-19, significantly more those who had IgG antibodies (p<0.05), while 23.3% had a positive attitude toward vaccination. ><05).”. This is not correct as the p-value refers to the different distribution across the three categories (no, I don’t know and yes) between the two groups.
  3. Table 2 does not report only vaccination status in relation to the attitudes towards vaccination because it also shows the result in relationship to the expected chance to get the COVID-19 in the absence of vaccination (that is not an attitude).
  4. Authors quoted “A significantly higher percentage of physicians received the Pfizer- BioNTech (Comirnaty) vaccine (Figure 1).”. This information should be reported earlier when the number of HCWs vaccinated is provided. Furthermore, are there differences only in respect to Pfizer vaccine? The figure is not clear as the p-value is reported at the bottom and does not refer to any of the three bar charts.
  5. Results about predictors of vaccination are messily reported. Looking at the table it can’t be concluded that age > 30 years was a positive predictor as the last class did not show a higher likelihood to be vaccinated. Furthermore, some variables that have been considered are not easy to understand. What does it mean SARS-CoV-2 PCR test performed? When? Why? The same refer to hospitalization. The confusion is also and mainly due to the fact that methods failed to report variables that were considered and how they were managed (and why).
  6. It is odd to not find the variable on attitudes in the multivariable analysis. It should appear. Actually, information reported in table 2 should be moved in table 3 for coherence.

Discussion

  1. Paragraph starting from line 174: repeated cross-sectional studies performed where and when?
  2. Lines 195-198: it is not willingness but uptake. Furthermore, people belonging to the 60-65 years of age group did not show a higher uptake of vaccination.

Conclusion

  1. Conclusion should be revised first because the paper did not focus on the increase of acceptance of vaccination and, second, because it misses to report how the study contributed to advance the knowledge in the field.

Author Response

Response to Reviewer 2 Comments

Reviewer #2: The paper “COVID-19 vaccination wailings and vaccine uptake among 2 healthcare workers: a single-center experience” is a single-center study that addresses HCW’s attitudes toward COVID-19 vaccination and following uptake investigating predictive factors. The paper provides a piece of information about the topic that adds to the existing evidence but need to be amended in respect to several aspects, in particular concerning the methodology. My comments are as follows.

Title

Point 1.There is a typo in the title (wailings instead of willingness).

Response 1: We apologize for making a typo in the title. We corrected this mistake and changed misspelled word into “willingness”.

Abstract

Point 2. The results about the two groups being compared in respect to attitudes and following uptake of vaccination are not clearly reported.

Response 2: Thank you for this suggestion. We changed that part of the results and we hope that it is now more clearly reported. following section into “HCWs who were hesitant about vaccination had almost two times higher odds (OR: 1.84; 95%CI: 1.04-3.25) concerning those with a negative attitude, while HCWs with a previous positive attitude had even 2.48 higher odds (OR:2.48; 95%CI:1.23-5.01) for vaccination.”

Introduction

Pont 3. The paragraph at lines 48-56 seems out of the scope of the paper. The authors should address the relevance of assessing vaccination attitudes and then describe the steps of the introduction of vaccination in Serbia.

Response 3: Thank you for this comment. We deleted paragraph at lines 48-49 and added three new paragraphs (paragraphs: 2-4) with additional information about vaccination attitudes and introduction of vaccination in Serbia.

Point 4. In the objective I would not mention the 9 months of follow up, but I would generically say that the objective was to assess the actual uptake of vaccination after its implementation.

Response 4: Thank you for this comment. We removed “9 months of follow up” from the objective.

Methods

Point 5. Authors said that all employees, including both medical and non-medical staff, were invited to participate to the study but, from results, it appears that only healthcare workers participated to the study. Indeed, it does not make sense to say that the sub-cohort of HCWs were involved in the follow up study. Please clarify this point.

Response 5: The reviewer is absolutely right in this point. We corrected in the Method section that "all health care workers, presently working were invited to participate”. Also, the term sub-cohort was misused. In fact, it is the cohort of HCWs that entered the seroprevalence study and that was followed for nine months. We have corrected this error in Methods.

Point 6. Who did administer the questionnaire to people involved in the study? The information should be reported in methods and not at the end of discussion.

Response 6 : Thank you for this valuable suggestion. We added this information in the Methods section. Besides that, we added the paragraph on the serological test used to determine antibody levels.

Point 7. The primary endpoint of the study is not clear. Indeed, it should be the willingness to vaccinate before the vaccination was available and the following uptake of vaccination.

Response 7: Thank you for this comment. We changed this endpoint as suggested.

Point 8. Methods and cut-off used to measure IG antibodies should be reported.

Response 8: We thank reviewer for this comment. We added in the Methods that “Determination of IgG antibodies against SARS-CoV-2 was performed using indirect ELISA SARS-CoV-2 IgG. Test was created at the Institute for Application for Nuclear Energy (INEP, Belgrade, Serbia), the CE labeled and registered as the IVD at the Serbian Agency for Drugs and Medical Devices (ALIMS). Results are expressed semi-quantitatively, as index, relative to reference calibrator sample. Cut off values for IgG were established based on ROC curve analysis. For antibody index higher than 20 the IgG test was considered positive. We then re-arranged that part of the method to make sense of the text.

Point 9. The source of data on vaccination status is not declared and source of information about RT-PCR testing and COVID-19 diagnosis are not clearly reported and not considered in the results.

Response 9: We thank the reviewer for those valuable comments. We have explained in detail the source of data on vaccination at the national level and at the hospital level (7th paragraph in the Methods section). Besides, we explained the functioning of the National electronic database on RT-PCR testing and the availability of this data on each tested HCWs.

Point 10. In the statistical analysis, I would suggest Authors to better explain which kind of analyses they performed in respect to the different endpoints.

Response 10 : Thank you for pointing this out. SARS-Cov-2 RT-PCR testing and COVID-19 diagnosis were used only in univariate and multivariable logistic regression to identify factors associated with the uptake of vaccine against COVID-19. We added this part in the Methods. We apologize for making a misunderstanding in the text. Therefore, we deleted part about the secondary endpoints.

Results

Point 11. Authors quoted “There was no statistically significant difference among IgG positive and IgG negative participants regarding age, gender, profession type, and years of service (Table 1).” Nevertheless, p-values were not reported in the table and methods section did not anticipate the use of test suitable for addressing the differences between the two groups in respect to all the variable considered that were not all categorical.

Response 11: We thank to the reviewer. We added the p value in the table 1. In the section Statistical analysis, we added that we used the independent t-test continuous variables.

Point 12. Authors quoted “almost half of the respondents (48.0%) were uncertain if they would like to be vaccinated against COVID-19, significantly more those who had IgG antibodies (p<0.05), while 23.3% had a positive attitude toward vaccination. ><05).”. This is not correct as the p-value refers to the different distribution across the three categories (no, I don’t know and yes) between the two groups.

Response 12: Thank you for this valuable comment. In table 1, we added p-value for comparison of two groups (with negative IgG and with positive IgG) and explained in the text that “Almost half of the respondents (48.0%) were uncertain if they would like to be vaccinated against COVID-19 while 23.3% had a positive attitude toward vaccination. Significantly more HCWs who had IgG antibodies than those without antibodies were uncertain about vaccination (62.5% vs. 49.0%).

Point 13. Table 2 does not report only vaccination status in relation to the attitudes towards vaccination because it also shows the result in relationship to the expected chance to get the COVID-19 in the absence of vaccination (that is not an attitude).

Response 13: Thank you for this mention. We changed table caption as advised.

Point 14. Authors quoted “A significantly higher percentage of physicians received the Pfizer- BioNTech (Comirnaty) vaccine (Figure 1).”. This information should be reported earlier when the number of HCWs vaccinated is provided. Furthermore, are there differences only in respect to Pfizer vaccine? The figure is not clear as the p-value is reported at the bottom and does not refer to any of the three bar charts.

Response 14: We agree with the reviewer’s comment. We calculated p values for every vaccine and presented them in the figure. We rearranged order in the Results and provided this information as you suggested.

Point 15. Results about predictors of vaccination are messily reported. Looking at the table it can’t be concluded that age > 30 years was a positive predictor as the last class did not show a higher likelihood to be vaccinated. Furthermore, some variables that have been considered are not easy to understand. What does it mean SARS-CoV-2 PCR test performed? When? Why? The same refer to hospitalization. The confusion is also and mainly due to the fact that methods failed to report variables that were considered and how they were managed (and why).

Respons15: Thank you for this valuable comment. We corrected and changed this paragraph as the following: “Predictors of healthcare workers' COVID-19 vaccine uptake are presented in table 3. In the univariate analysis, COVID-19 vaccine uptake was associated with female sex, age 30-59 years, years of service > 20 years, SARS-CoV-2 PCR test performed, hospitalization, hesitant and positive attitude about vaccination, while nurses were less likely to get vaccinated. Multivariate analysis shows that females would more likely get vaccinated than males. Age 30-59 years were associated with higher uptake of COVID-19 vaccine. Nurses were less likely to get vaccinated, confirming the results from the univariate analysis. After multivariate analysis HCWs who were hesitant about vaccination had almost two times higher odds (OR: 1.84; 95%CI: 1.04-3.25) concerning those with a negative attitude, while HCWS with a previous positive attitude had even 2.48 higher odds (OR:2.48; 95%CI:1.23-5.01) for vaccination.” Besides, we added in the Methods section the information about RT-PCR testing and hospitalization and how they were managed.

Point 16. It is odd to not find the variable on attitudes in the multivariable analysis. It should appear. Actually, information reported in table 2 should be moved in table 3 for coherence.

Response 16: We thank the reviewer for these suggestions. We moved OR (95% CI) values from table 2 to table 3 and included variable on attitudes in the multivariable analysis. With this model SARS-CoV-2 PCR testing and hospitalization didn’t show significant in predicting vaccine uptake as was in previous model but attitude on vaccination was significant instead.

Discussion

Point 17. Paragraph starting from line 174: repeated cross-sectional studies performed where and when?

Response17: Thank you. We added data about where and when are these cross-sectional studies performed.

Point 18. Lines 195-198: it is not willingness but uptake. Furthermore, people belonging to the 60-65 years of age group did not show a higher uptake of vaccination.

Response 18: Thank you for this comment. We changed into “… uptake of vaccine among HCWs was associated with age groups 30-59 years.”

Conclusion

Point 19. Conclusion should be revised first because the paper did not focus on the increase of acceptance of vaccination and, second, because it misses to report how the study contributed to advance the knowledge in the field.

Response 19: We revised the Conclusion and added that “The results of this and many other similar studies are important as they can affect future willingness rate of HCWs and overall population to get vaccinated against COVID-19.”

Round 2

Reviewer 1 Report

I appreciate the effort of the authors to improve/strengthen the manuscript. Although there may be some minor English usage issues, the manuscript is clearer.

The additional material is also useful in highlighting the significance of the study and the results.

Author Response

Reviewer #1:

Comments and Suggestions for Authors

I appreciate the effort of the authors to improve/strengthen the manuscript. Although there may be some minor English usage issues, the manuscript is clearer.

The additional material is also useful in highlighting the significance of the study and the results.

Response:  We  would  like  to  thank  Reviewer  for  taking  the  time  and  effort  necessary  to  review the  manuscript.  We  sincerely  appreciate  all  valuable  comments  and  suggestions, which  helped us  to  improve  the quality  of the manuscript.

Reviewer 2 Report

The paper has been improved but still lacks clarity.

Abstract

  1. I would recommend further revising the results. It is not clear at all for readers when the Authors report that previous vaccine-hesitant and those with previous vaccine positive attitudes were both more likely to vaccinate. See also my comments below.

Introduction

  1. Authors have added the following sentence “It was shown that the attitude of colleagues towards vaccination and trustful relationships with them have a more significant role in changing the decision to vaccinate than the influence of mass media.” This information is in my opinion not useful for the purpose of the study as the Authors did not explore further these aspects in their analysis
  2. The objective of the paper was also to investigate factors associated to vaccine uptake.

Methods

  1. I am still baffled by the management of variables SARS-CoV-2 PCR test positive (ever)and hospitalization as I guess that this information were only available for people being tested and with COVID-19. Indeed, you had missing data for the rest of the sample. How did you deal with it in the model? I think that a subgroup analysis should be better.
  2. The management of the information on HCWs’ willingness to vaccinate should be better dealt with (see also my comments below).

Results

  1. Table 1 has an asterisk (that I guess refers to missing data) but the caption does not report any information about its meaning. Furthermore, missing data, if any, should be reported.
  2. Author quoted “Significantly more HCWs who had IgG antibodies than those without antibodies were uncertain about vaccination (62.5% vs. 49.0%).” Again, the difference is across the three groups. There were also less people answering “no” among those IgG positive and this is also expected because it was not known at the beginning if vaccination was necessary for positive patients. This aspect has not been considered in the paper.
  3. In my opinion, the results about the type of administered vaccine are misleadingly reported. The comparison should be done across professionals in respect to received vaccines. Now the comparison is done among professionals for each vaccine. Authors could also leave things as they are but should revise the sentence at lines 176-178.
  4. Table 2 does not make sense. Why every row has a p-value? You should compare the three groups in respect to vaccination and have only one p-value. Please revise it. I would suggest finding a unique method to deal with the variable avoiding the use of confusing term. Hesitancy include refuse but also delay. In my opinion both people answering “no” and “I do not know” could be considered hesitant. Please find a way to report results in a straightforward way.

Discussion

  1. The sentence on physicians/doctors’ (please use the same words throughout the text) preference for mRNA vaccines should be revised according to my comment before.

Author Response

Reviewer #2:

Abstract

  1. I would recommend further revising the results. It is not clear at all for readers when the Authors report that previous vaccine-hesitant and those with previous vaccine positive attitudes were both more likely to vaccinate. See also my comments below.

Response: We thank the reviewer for the helpful suggestion. We made additional analysis (added in response 5 and changed sentences in the Abstract as follows:

“Before vaccines became available, among IgG-positive HCWs compared to IgG-negative, a significantly higher percentage of HCWs was unsure whether to be vaccinated (62.5% vs. 49.0%), and significantly less stated that they would not be vaccinated 16.7% vs. 25.1%). However, when the vaccines became available in our country, a total of 64.3% of these two groups changed their attitude and received the vaccine during the first 9 months of vaccination (53.6% of those previously unsure and 19.1% of previously without the intention of being vaccinated).”

Introduction

  1. Authors have added the following sentence “It was shown that the attitude of colleagues towards vaccination and trustful relationships with them have a more significant role in changing the decision to vaccinate than the influence of mass media.” This information is in my opinion not useful for the purpose of the study as the Authors did not explore further these aspects in their analysis

Response:  The reviewer is right. We recognized that this sentence did not fit into the context; we deleted it. We added new reference No 12.

  1. The objective of the paper was also to investigate factors associated to vaccine uptake.

            Response:  We agree with the reviewer. We have added the suggested part to the objective of the paper and changed the objectives to  ”The aim of this study was: a) to assess the intention of vaccination against COVID-19 before the vaccines were approved; b) to assess the actual uptake of vaccination c) to investigate factors associated with vaccine uptake after its implementation among HCWs”

Methods

  1. I am still baffled by the management of variables SARS-CoV-2 PCR test positive (ever)and hospitalization as I guess that this information were only available for people being tested and with COVID-19. Indeed, you had missing data for the rest of the sample. How did you deal with it in the model? I think that a subgroup analysis should be better.

Response:  We followed a cohort of HCWs from a cross-sectional study (conducted in June 2020) until the end of September 2021. Indeed, it is not accurate to say that they were ever tested, but during the follow-up period. We changed that in the table e.i we deleted “ever”.

But we did not have any missing information regarding testing information (data are dichotomous: tested/not tested). In addition, we had no missing value for hospitalized health care workers. Hospital Occupational Health Department keeps records of all hospitalized HCWs. There were a total of 20 of them, all hospitalized at our hospital, e.i. where they are working. During the COVID-19 outbreak, the surveillance of all HCWs is organized by Infection Control Team too. Sick HCWs themselves or their relatives have the obligation to contact the Occupational Health Department to justify the illness. The Infection Control Team contacts them for further surveillance details. Thus the variable “hospitalization” is dichotomous (Yes/No) for all HCWs. We added this in the Method section.

  1. The management of the information on HCWs’ willingness to vaccinate should be better dealt with (see also my comments below).

 Response:  In line with the reviewer’s suggestion, we tried to present the subgroups differently, ie. to form only 2 groups: hesitant (who refuse but also who delay) as reviewer’s comment No 9, and the second group of vaccination willingness. Before that, we compared 2 subgroups in the hesitant group.

Vaccine hesitance

Vaccination

P value

NO

n (%)

YES

n (%)

No

59 (46.5)

68 (53.5)

0.002

Unsure

85 (30.8)

191 (69.2)

Total

144

259

Out of a total of 403 such persons who were hesitant at the beginning of the study, 259 (64.3%) received vaccine. The difference was statistically significant, p = 0.002: among vaccine-unsure there were much more vaccinated (191 69.2%) then non-vaccinated (85 30.8%) while among those who were against the vaccination 68, 53.5% were vaccinated. Since the new two groups were not homogeneous, this means that we can observe them separately and we decided to leave 3 categories (No/Unsure/Yes), as it was presented in Tables 2 and 3. We believe that the most important conclusion in our study is a significant change of mind about vaccination among unsure HCWs.

Results

  1. Table 1 has an asterisk (that I guess refers to missing data) but the caption does not report any information about its meaning. Furthermore, missing data, if any, should be reported.

Response: We apologize for this mistake. The asterisk remained from the first submitted version of the manuscript, with which we marked the statistical significance (listed below the table). We forgot to delete it in the corrected version of Table 1 when we added a p-value for each variable.

  1. Author quoted “Significantly more HCWs who had IgG antibodies than those without antibodies were uncertain about vaccination (62.5% vs. 49.0%).” Again, the difference is across the three groups. There were also less people answering “no” among those IgG positive and this is also expected because it was not known at the beginning if vaccination was necessary for positive patients. This aspect has not been considered in the paper.

Response: We appreciate the reviewer’s comment and agree with it. In order to present the result more clearly, we have changed that part in the result section as follows:

“Among IgG-positive HCWs compared to IgG-negative HCWs, there was a statistically a significantly higher percentage of those who were unsure about vaccination, and a lower percentage of those who answered that they did not want to be vaccinated (62.5% vs. 49.0% and 16.7% vs. 25.1%, respectively) when vaccines will be available.”

  1. In my opinion, the results about the type of administered vaccine are misleadingly reported. The comparison should be done across professionals in respect to received vaccines. Now the comparison is done among professionals for each vaccine. Authors could also leave things as they are but should revise the sentence at lines 176-178.

Response: We thank the reviewer for this valuable comment. We re-designed Figure 1 and presented in a more precise way that a significantly higher percentage of physicians received the Pfizer vaccine than the other two vaccines. We changed the sentence at lines 176-178.

  1. Table 2 does not make sense. Why every row has a p-value? You should compare the three groups in respect to vaccination and have only one p-value. Please revise it. I would suggest finding a unique method to deal with the variable avoiding the use of confusing term. Hesitancy include refuse but also delay. In my opinion both people answering “no” and “I do not know” could be considered hesitant. Please find a way to report results in a straightforward way.

Thank you for this important suggestion. Indeed, we compared the three groups (HCWs who expressed willingness to be vaccinated, i.e. with the answer Yes, those who were against vaccination, i.e. with the answer No, and HCWs with the response I don’t know-unsure. We replaced the term hesitant with unsure. In response to the reviewer's comment No. 5, we explained in detail why we did not merge 2 groups (No / Unsure) into one (Hesitant). Therefore, we changed Table 2 and added one p-value for each variable. Although there are published articles that present together people who answered No and I don’t know-unsure and marked them as hesitant, we decided to keep three categories of answers, because in this way, it is clear that HCWs who were unsure were significantly changed their attitude when the vaccine became available, and a large percentage of them received the vaccine. We also explained additional analysis in our responses 4 and 5.

Discussion

  1. The sentence on physicians/doctors’ (please use the same words throughout the text)

       preference for mRNA vaccines should be revised according to my comment before.

Response: We checked through the entire text and changed the term “doctors” into physicians (in Table 1 and 2). The sentence in the discussion is in line with the changed schedule (Physicians preferred messenger RNA (mRNAvaccine probably due to a greater knowledge of the new manufacturing technology, compared to other hospital staff). We also added the sentence: “The results of our survey are in line with the results of the above study.”

Round 3

Reviewer 2 Report

Abstract

The sentence that the Authors have included is confusing as in in the first part they distinguished between IgG-positive and IgG-negative HCWs whereas in the second part they reported crude numbers.

Methods

The problem of variable management is still there. I am aware of the type of variables but hospitalization as well as positivity to PCR can be recognized only in people who have got the infection and have submitted to testing. This is why I suggested to have a subgroup analysis on them.

Another aspect that still need to be addressed is the reporting of how the main variables of the study were managed. I understand why the Authors have left the distinction among people unsure and not willing to vaccinate but Authors should clearly report this choice in methods.

Results

In my opinion, the results about the type of administered vaccine are still misleadingly reported. If you want to represent if the type of vaccine depends on profession a four by three comparison should be performed. It does not make sense to assess if differences exist among health professionals for each vaccine.

Author Response

Response to reviewer # 2

We thank the reviewer for their careful reading of the manuscript and their constructive remarks. We have taken the comments on board to improve and clarify the manuscript. Please find below a detailed point-by-point response to all comments

Abstract

The sentence that the Authors have included is confusing as in the first part they distinguished between IgG-positive and IgG-negative HCWs whereas in the second part they reported crude numbers.

Response: We agree with the reviewer’s comment. We made an additional analysis of vaccinated HCWs according to seroprevalence as it was done in the analysis about the attitude toward vaccination before vaccines were available in our country. Instead of the previous sentence, we added a new one:  “When the vaccines became available in our country, among IgG-positive HCWs, there was no statistical difference in vaccine uptake regarding those with a previous negative, positive and unsure opinion about vaccination (p=0.498). Among IgG-negative HCWs, a significantly higher percent of previously unsure individuals received some of the available vaccines (51.4% vs.44.4%), and a lower percentage of those with a negative attitude toward vaccination previously (20.3% vs. 34.0%).”

Besides that, we also changed Table 2 to add additional statistical analysis. We hope that the presentation of the results in this way will be much clearer to the readers.We thank the reviewer once again for this valuable help.

Methods

The problem of variable management is still there. I am aware of the type of variables but hospitalization as well as positivity to PCR can be recognized only in people who have got the infection and have submitted to testing. This is why I suggested to have a subgroup analysis on them.

Response: We agree with the reviewer’s comment. We agree that PCR positives can be only to those who were tested, and hospitalized only those who were COVID-19 ill. Accordingly, we formed a new variable, “History of COVID-19” (Table 3), in which HCWs were divided into 3 subgroups: those who were PCR negative, those who were PCR positive but who were not hospitalized, and those PCR positive who were hospitalized. No one HCW was hospitalized without a positive PCR test. The difference in the view of vaccination was not statistically significant, nor were there any changes in the multivariate analysis.

Another aspect that still need to be addressed is the reporting of how the main variables of the study were managed. I understand why the Authors have left the distinction among people unsure and not willing to vaccinate but Authors should clearly report this choice in methods.

Response: We thank the reviewer for this valuable comment. We have added an explanation of these variables in the Method section. 

Results

In my opinion, the results about the type of administered vaccine are still misleadingly reported. If you want to represent if the type of vaccine depends on profession a four by three comparison should be performed. It does not make sense to assess if differences exist among health professionals for each vaccine.

Response: We thank a reviewer for this important suggestion. We performed a new statistical analysis and changed Figure 1, and added that “A significantly higher percentage of nurses received the Sputnik vaccine, while the same percentage of nurses and physicians received the Pfizer vaccine.“In addition, we added one sentence in the Discussion.

Round 4

Reviewer 2 Report

Dear authors, there are still problems in the reporting of data. When  you refer to the uptake of vaccination among unsure HCWs the counterfactual situation is represented by those who had a positive attitudes. This is not the information that is conveyed in the sentences you have included. Again there are some problems in the evaluation of vaccine types among HCWs. If you look at the figure it is not easy to understand which are the denominators. Percentages in single HCW group do not sum up to 100%. Please submit the paper to statistical review.  

Author Response

Response to reviewer # 2

Dear authors, there are still problems in the reporting of data. When you refer to the uptake of vaccination among unsure HCWs the counterfactual situation is represented by those who had a positive attitudes. This is not the information that is conveyed in the sentences you have included.

Response: We thank the reviewer for this additional comment. We agree that the data on the percentage of vaccinated among those who had a positive attitude towards immunization was not highlighted. We changed the sentence to the following in the Results: “Among IgG-negative HCWs, those who stated clear positive (yes) and clear negative (no) attitude toward vaccination, before the immunization period had begun, were vaccinated in only 28% and 20%, respectively, while unsure HCWs got a vaccine in even 51% (p=0.006). There was no significant difference in the vaccination rate according to the previous attitude toward vaccination among IgG-positive HCWs (p=0.498).” Also, we changed this part in the Abstract.

Again there are some problems in the evaluation of vaccine types among HCWs. If you look at the figure it is not easy to understand which are the denominators. Percentages in single HCW group do not sum up to 100%. Please submit the paper to statistical review. 

Response: To be more clear and in accordance with the reviewer's concerns, after consultation with a statistician, and according to the reviewer's suggestion, we changed the presentation of results in Figure 1: the sum of administrated vaccines is 100% for each HCW group now.

Accordingly, we changed the previous sentence in the Results as: “A significantly higher percentage of physicians received the Pfizer- BioNTech (Co-mirnaty) vaccine than nurses and other staff, while more other staff received Sinofarm (BBIBP-CorV) vaccine compared to nurses and physicians.”

Also, we changed the sentence in the Discussion as: “Other HCWs have greater confidence in vaccines made using traditional technology such the inactivated whole virus vaccines”.

We hope we have responded correctly to all of the reviewer's suggestions now.
